# Unifying information theory and machine learning in a model of electrode discrimination in cochlear implants

Xiao Gao [1,2]*, David Grayden[1], Mark McDonnell[3]

**1** Department of Biomedical Engineering, University of Melbourne, Parkville, VIC, Australia, **2** School of Physics, The University of Sydney, Sydney, NSW, Australia, **3** Computational Learning Systems Laboratory, School of Information Technology & Mathematical Sciences, University of South Australia, Mawson Lakes, SA, Australia

* xiao.gao@unimelb.edu.au

**Data Availability Statement:** All relevant data are within the manuscript and its Supporting information files.

## Abstract

Despite the development and success of cochlear implants over several decades, wide inter-subject variability in speech perception is reported. This suggests that cochlear implant user-dependent factors limit speech perception at the individual level. Clinical studies have demonstrated the importance of the number, placement, and insertion depths of electrodes on speech recognition abilities. However, these do not account for all inter-subject variability and to what extent these factors affect speech recognition abilities has not been studied. In this paper, an information theoretic method and machine learning technique are unified in a model to investigate the extent to which key factors limit cochlear implant electrode discrimination. The framework uses a neural network classifier to predict which electrode is stimulated for a given simulated activation pattern of the auditory nerve, and mutual information is then estimated between the actual stimulated electrode and predicted ones. We also investigate how and to what extent the choices of parameters affect the performance of the model. The advantages of this framework include i) electrode discrimination ability is quantified using information theory, ii) it provides a flexible framework that may be used to investigate the key factors that limit the performance of cochlear implant users, and iii) it provides insights for future modeling studies of other types of neural prostheses.

## Introduction

People with hearing loss have greatly benefited from continuously developing cochlear implant technology. Contemporary cochlear implants restore functional hearing by directly stimulating different locations in the cochlea with electrodes [1]. The tonotopic organization of surviving nerve fibers allows cochlear implant electrodes to stimulate distinct neural populations thereby coding spectral information to electrode positions [2]. However, there is wide variability of speech perception performance between cochlear implant users [3–5]. The main factors that are known to limit speech recognition include i) age at implantation, age of onset of

**Funding:** XG receives a McKenzie Fellowship from the University of Melbourne (https://sites.research.unimelb.edu.au/research-funding/researcher-development-schemes/mckenzie-fellowship). The funders had no role in study design, data collection and analysis, decision to publish, or preparation of the manuscript.

**Competing interests:** The authors have declared that no competing interests exist.

deafness, and survival rate of auditory nerve fibers [6], ii) electrode placement and insertion depth [7, 8], and iii) channel interactions between electrodes [3, 9].

Electrode discrimination has been shown to correlate with performance of speech tasks [10, 11]. It has been used as a one of the psychophysical measurements to assess pitch ranking ability [5, 12–14], to investigate electrode stimulation strategies [13], to evaluate the effects of electrode placement and insertion [11, 15, 16], and to determine the number of spectral channels that can be distinguished by cochlear implant users [7]. In particular [11, 15], reported that electrode distance to the inner wall of the cochlea is a significant variable for the prediction of electrode discrimination ability, and improvements in speech recognition are expected with electrode placements closer to the inner wall [13, 17]. Showed that the Dual-Electrode strategy achieves better performance in tonotopic pitch ranking compared to single-electrode strategy. However, the results varied among cochlear implant users [12, 15, 16], so the extent to which the key factors affect the performance of cochlear implants users is still unknown.

Mathematical and computational models have been developed to study electrically evoked auditory nerve activity [18–20], sound coding strategies [21], electrical current spread in the cochlea [22], electrode stimulation strategies [23–25], and effects of electrode placements and insertion depth [26, 27]. The computational models are used to evaluate the effects of different electrode configurations and to study the pattern of neural excitation produced by a given electrode configuration. Psychophysical cochlear implant models which study the psychophysical responses to cochlear implant stimulation, on the other hand, have incorporated neural models to evaluate hearing performance [28]. Proposed an analytical model to predict loudness growth for different electrode configurations [29, 30]. Developed a computational model of pitch perception and discussed the contribution of place and temporal cues to pitch perception [31, 32]. Demonstrated how the spatial spread of the stimulation field impacts on speech recognition and predicted speech intelligibility. More recently [33], studied the effects of musical training on fundamental frequency discrimination.

However, to enable modelling studies of speech recognition variability at the individual level, a modelling framework is needed where the geometric information of the cochlea, the electric conduction characteristics and the neural elements are individually fitted. Our previous studies modeled the electrode-to-fiber interface and quantified the performance of electrode place discrimination from a theoretical point of view [34–38]. This modeling framework was developed based on the place code theory of hearing [39], which assumes that sound frequency perception depends upon different auditory nerve fibers responding to different frequencies of sound. Accordingly, the interface between the array of electrodes and the auditory nerve fibers is conceptualized as a communication channel, where the channel input random variable is defined as a choice of electrode to stimulate, and the channel output random variable is a function of the nerve fibers that respond to the electrode stimulation. The mutual information between channel input random variables and channel output random variables is assumed to act as a proxy for electrode discrimination ability, under the assumption that increased mutual information leads to increased hearing performance of cochlear implants [40]. In later studies [41, 42], the performance of place discrimination was quantified by predicting the location of the stimulating electrode in response to an activation pattern of the fibers, and a commonly used measurement of electrode discrimination, four-interval forced-choice, was simulated by applying a discriminative classifier proposed in [43, 44]. However, the performance of cochlear implant electrode discrimination ability has only been investigated based on the statistical correlations [34, 35] or predicted using a neural network classifier [41, 42]. The methods have not been unified to address clinically relevant issues.

The goal of the current study is to provide a flexible structure that enables the investigation of the extent to which key factors limit the performance of cochlear implants model. Unifying

the information theoretic method and the neural network classifier into a modeling framework enables us to quantitatively interrelate the performance of cochlear implants model to electrode discrimination ability. In particular, we aim to investigate the extent to which the key model parameters affect the modeled electrode discrimination ability. These parameters—electrode placements, insertion depth, auditory nerve fibre survival rate, width of current spread—represent the key factors that are thought to affect hearing performance with cochlear implants. Therefore, this study may provide insight into the following important and clinically relevant questions: i) how to quantitatively analyze the variability in hearing performance of cochlear implants at the individual level, ii) whether more electrodes can provide improved hearing performance, and iii) how to personalize cochlear implant configurations for an individual user.

## Methods

### Generation of nerve fiber activation pattern

Our previously developed model is adapted as the underlying method to generate fiber activation patterns [34, 42], which we summarize here:

1. geometry of auditory nerve fiber and electrode locations and quantities;

2. mechanisms of stochastic action potential generation in auditory nerve fibers;

3. current spread from each electrode;

4. dependence of loudness perception on overall auditory nerve activity;

We first describe the locations and distances between electrodes and auditory nerve fibers. The locations of the $i$–th fiber, $f_i$, and $j$–th electrode, $e_j$, are defined in three-dimensional space as [45]

$$f_i = (R_{f,i}\cos(\varphi_{f,i}), R_{f,i}\sin(\varphi_{f,i}), h_{f,i}), \tag{1}$$

$$e_j = (R_{e,j}\cos(\varphi_{e,j}), R_{e,j}\sin(\varphi_{e,j}), h_{e,j}), \tag{2}$$

where $i = 1, \ldots, N$, $N$ is the number of fibers, $j = 1, \ldots, M$, and $M$ is the number of electrodes. In Eq (1), $\phi_{f,i}$ is the twirling angle of the $i$–th fiber and $R_{f,i}$ and $h_{f,i}$ are the radius distance and the height of the $i$–th fiber, respectively. Similarly, in Eq (2), $\phi_{e,j}$, $R_{e,j}$, and $h_{e,j}$ define the quantities for the $j$–th electrode. An example electrode-to-fiber geometry is shown in Fig 1.

The second component of the underlying model defines the event of a fiber producing an action potential under a memoryless assumption. Let $Y_i \in \{0, 1\}$ denote whether or not fiber $i$ produces an action potential in response to current at fibre $i$, $C_{f,i}$. The probability that $Y_i = 1$, given $C_{f,i}$ is

$$P_s(C_{f,i}) := 0.5\left(1 + \text{erf}\left(\frac{C_{f,i} - \theta_i}{\sqrt{2}\sigma_i}\right)\right), \tag{3}$$

where $\theta_i$ represents the current that produces an action potential with probability 0.5 and $\sigma_i$ is a parameter that is used to measure the probability of an action potential [22, 34].

The third component defines the current level for the $i$–th fiber, $C_{f,i}$, as a function of the current level, $C_{e,j}$, at stimulating electrode, $j$, as

$$C_{i,j} = C_{e,j}10^{-0.05Ad_{i,j}l}, \tag{4}$$

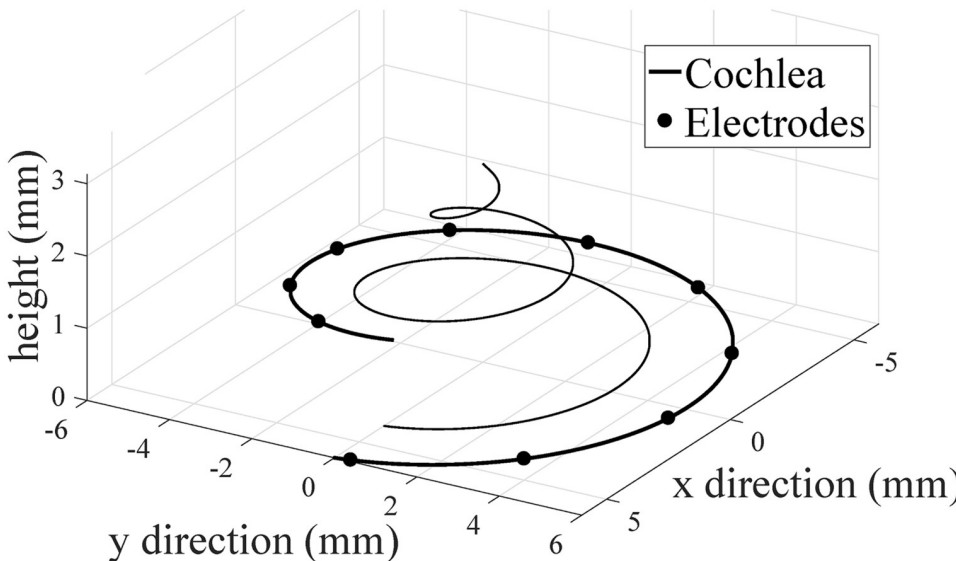

**Fig 1. The geometry of the electrode array and auditory nerve.** In this example, the total twirling angle of the cochlea is $5\pi$ radians, the twirling angle of the electrode array is $2\pi$ radians, and the distance between the electrodes and auditory nerve fibers is $r = 2$ mm. This figure is adapted from Figure 1 in [42].

where $d_{i,j}$ is the distance between the $i$–th fiber and the $j$–th electrode, $l$ is the total length of the cochlea, and $A$ is the attenuation in electrode current [46]. The results in [46] confirm the attenuation is approximately 0.5 dB/mm for monopolar and 4 dB/mm for bipolar stimulation. In this paper, we follow [46] and use a single biphasic pulse stimulus and choose $A$ between 0.5 dB/mm and 4 dB/mm to model the different widths of current spread. We use this simple yet effective model for action potential generation and current spread. More complex models may improve the accuracy of modeling electrode stimulation strategies [28].

The fourth component models the loudness by assigning a current level to each electrode such that stimulation of each electrode causes no more than a certain percentage of the auditory nerve fibers to generate action potentials.

A vector of the outcomes for a total of $N$ nerve fibers is defined as the channel output in response to an electrode choice, $e_j$,

$$\mathbf{Y} = (Y_1, Y_2, \ldots, Y_N). \tag{5}$$

Since $Y_i$ is the activation of the $i$–th fiber, $\mathbf{Y}$ is the total fiber activation pattern in response to an electrode choice.

Fig 2 shows three examples of activation patterns of auditory nerve fibers, $Y$. The electrode-cochlea geometry used is that in Fig 1, $N = 1000$, and $A = 0.5$ dB/mm. For a given stimulated electrode, most of the action potentials were generated in the fibers closest to the stimulated electrode. The activation patterns generated by the model were used to quantify the electrode discrimination ability.

## Quantifying electrode discrimination ability

The method introduced by [44] is adapted to classify the fiber activation pattern. To this end, each example pattern, $\mathbf{Y}$, is assigned to one of the $M$ electrodes. The performance of electrode discrimination is then estimated by calculating the correct classification rates by simulating many repeated stimulations of each electrode using the model.

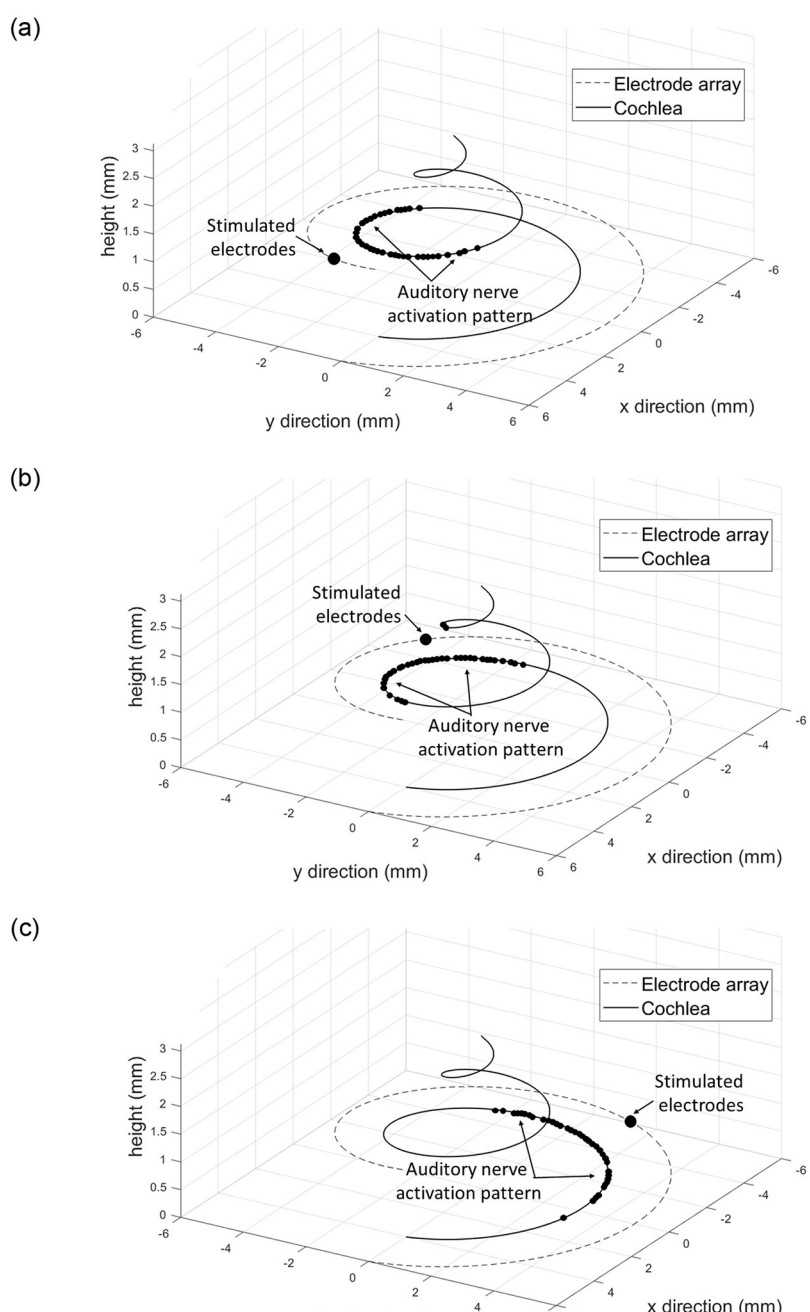

**Fig 2. The activation pattern of auditory nerve fibers, *Y*, for a stimulated electrode.** Three example cases are shown. The electrode-cochlea geometry is that shown in Fig 1, *N* = 1000, and *A* = 0.5 dB/mm.

The classifier, used by [42], consists of a single hidden layer feedforward network (SLFN), where *N* is the number of units in the input layer (equal to the number of fibers), *L* is the number of units in the hidden layer, and *M* is the number of output layer units (equal to the number of electrodes). We also define two matrices that map the fiber activation samples (the input vectors) onto the indices of electrodes (the prediction vectors). The input weight matrix $\mathbf{W}_{\text{in}}$ (of size $L \times N$) maps the data from the *N*-dimensional input space to the *L*-dimensional

hidden-layer feature space; the output weight matrix $\mathbf{W}_{\text{out}}$ (of size $M \times L$) maps the $L$-dimensional hidden-layer activations to $M$-dimensional prediction vectors. The input and output weights are initially randomly chosen from uniform distributions on the interval $[-1, 1]$.

In the training phase, we define a matrix, $\mathbf{T}$, where each column represents an activation pattern, $\mathbf{Y}$. We denote $\mathbf{V}$, which is the same size as $\mathbf{T}$, as the labels of each training sample. Each column in $\mathbf{V}$ has only one non-zero element set to 1 as the label of the corresponding training vector in $\mathbf{T}$. We define $\mathbf{Z}$ as the prediction vectors

$$\mathbf{Z} = \mathbf{W}_{\text{out}}A_{\text{train}}, \tag{6}$$

where

$$A_{\text{train}} = f(\mathbf{W}_{\text{in}}\mathbf{T}). \tag{7}$$

In Eq (7), $\mathbf{W}_{\text{in}}\mathbf{T}$ are the hidden layer activations and $f(\cdot)$ is the termwise $(L \times P_1 \rightarrow L \times P_1)$ activation function. The matrix $\mathbf{W}_{\text{out}}$ is found in the training phase that minimizes the mean square error between $\mathbf{V}$ and $\mathbf{Z}$. In this paper, the absolute value function is used as activation function.

Similar to the training phase, we define $\mathbf{T}'$ as the test vector, where each column represents a fiber activation pattern. The prediction vector in the test phase is defined as

$$\mathbf{Z}' = \mathbf{W}_{\text{out}}f(\mathbf{W}_{\text{in}}\mathbf{T}'), \tag{8}$$

where $\mathbf{W}_{\text{out}}$ was obtained in the training phase. The classification decision for testing vector $i$ is made from $\mathbf{Z}'$ by choosing the maximum element in a given column. This corresponds to making a decision of which particular electrode was stimulated in response to a given fiber spiking pattern.

We note that the structure of the network used in this paper was introduced by [47] as a variation of the Extreme Learning Machine algorithm. The weights from the input to the hidden layer neurons are randomly initialized and are fixed thereafter (i.e., they are not trained). Output weights $\mathbf{W}_{\text{out}}$ are obtained by minimizing the mean square error between $\mathbf{V}$ and $\mathbf{Z}$. This is obtained using the closed form expression for such linear regression problems,

$$\mathbf{W}_{\text{out}} = \mathbf{V}\mathbf{A}_{\text{train}}^{\top}(\mathbf{A}_{\text{train}}\mathbf{A}_{\text{train}}^{\top})^{-1}. \tag{9}$$

We chose this network structure because of its low training complexity.

## Unifying information theory and machine learning in the framework for electrode discrimination

In [34], only mutual information was calculated to predict optimal number of electrodes, whereas in [42], only the correct classification rates are calculated to quantify electrode discrimination ability. To unify information theoretic methods and machine learning and investigate how mutual information and correct classification rates are related, in this paper, we model electrode discrimination using framework is adapted from [34, 42]. Correct classification rates were calculated between the stimulated and predicted electrodes and the mutual information between the inputs and outputs of the system were also estimated, as shown in Fig 3.

We write a vector $\mathbf{E} \in \{e_j: j = 1, \ldots, M\}$ that defines the stimulated electrode and $\mathbf{E}' \in \{e'_j : j = 1, \ldots, M\}$ that defines the predicted electrode in response to an actual stimulated electrode. We then estimate the probability of the system input, $P_E$, and output, $P_{E'}$, as well as the conditional probability, $P_{E'|E}$. Thus, the mutual information between the system

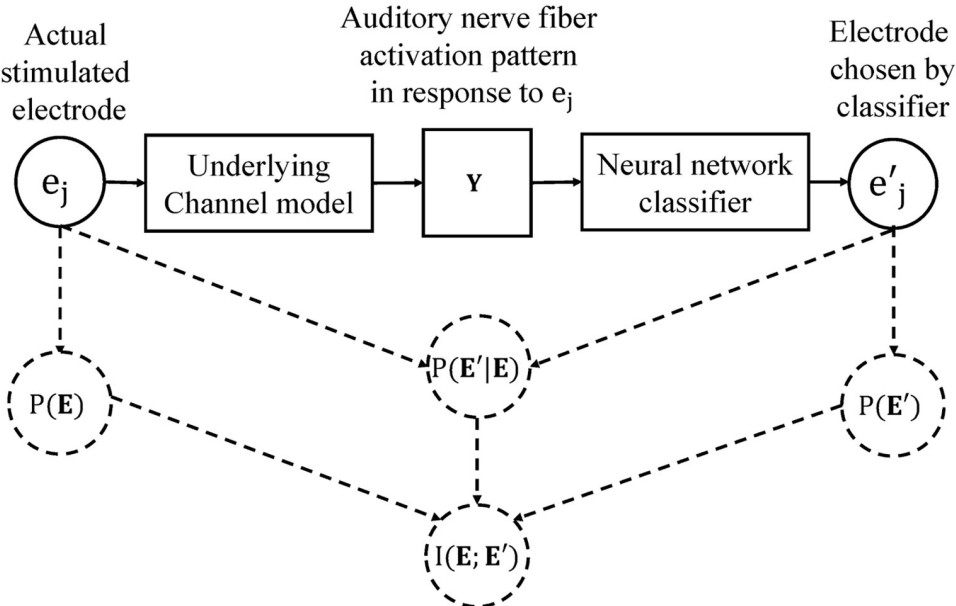

**Fig 3. A full framework for estimating electrode discrimination ability.** The probability of the system input, $P_E$, and output, $P_{E'}$, as well as the conditional probability, $P_{E'|E}$, are estimated after the testing phase of the neural network classifier. Mutual information is then calculated using Eq (11).

input and output is calculated as

$$I(\mathbf{E}; \mathbf{E}') = H(\mathbf{E}') - H(\mathbf{E}'|\mathbf{E}) \tag{10}$$

$$
\begin{aligned}
&= -\sum_{j=1}^{M} P_{E'}(j) \log_2 P_{E'}(j) \\
&+ \sum_{k=1}^{M} P_E(k) \times \sum_{j=1}^{M} P_{E'|E}(j|k) \log_2 \left( P_{E'|E}(j|k) \right).
\end{aligned}
\tag{11}
$$

## Parameter selection

Our framework provides a flexible structure that enables us to model key factors that are thought to limit the performance of cochlear implant model. The factors considered are listed in Table 1.

The surviving auditory nerve fibers are chosen as nonuniformly located along the cochlea according to a model of the actual location distribution (for details, see [37]). The attenuation

**Table 1. Parameters in the modeling framework.**

| Parameters | Description |
| --- | --- |
| $N$ | Number of surviving fibers. |
| $A$ | Attenuation in electrode current, which is used to model width of current spread. |
| $C_e$ | The value of current at a stimulated electrode, which is used to model loudness perception. |
| $\alpha$ | Electrode insertion depth, which is used to model different types of electrode array. |
| $r$ | Electrode-to-fiber distance, which together with $\alpha$ is used to model electrode placement. |

in current, $A$, is assumed to increase linearly with the distance from an electrode and the value can vary depending on the electrode stimulation strategy [48]. The interface between electrodes and auditory nerve fibers is modeled by electrode insertion depth, $\alpha$, and electrode-to-fiber distance, $r$, which enables us to freely adjust the geometry between electrodes and auditory nerve fibers based on clinical studies.

## Results

We quantify the performance of cochlear implant model by investigating the correct classification of electrode discrimination and estimating mutual information between the system inputs and outputs. The parameters and the training phase for the classifier follows the same principle used by [42]. In the test phase, for of each possible electrode location, the correct classification rate and mutual information are estimated from 400 tests.

### Correct classification rate vs. mutual information

Fig 4(a) and 4(b) show the mutual information and correct classification rates, respectively, for $\alpha = \{2\pi, 3\pi\}$. The rest of the parameters in the model are chosen as $N = 3000$, $A = 0.5$ dB/mm, and $r = 2$ mm. We emphasize that, in this paper, we aim to verify the method so the parameters are chosen to model a general case; the parameters can be chosen based on clinical studies to more accurately model cochlear implants of individual users. In Fig 4(a), for a given $M$, we see higher mutual information for $\alpha = 3\pi$, which is consistent with findings in previous modeling studies [34, 35]. Accordingly, in Fig 4(b), we find higher correct classification rate for a longer electrode array ($\alpha = 3\pi$). We observe a rapid increase of mutual information for very small number of electrode from 5 electrodes to around 11 electrodes; however, more than 11 electrodes only leads to a slight change in mutual information, which qualitatively agrees with the results of [42].

At the point where the peak of mutual information appears, we find that the correct electrode discrimination rate for both cases of $\alpha$ is around 90%. More than 11 electrodes leads to a sharp drop of the total correct classification rate. We note that the drop of correct classification rate is due to the distance between electrodes, which decreases with increasing numbers of electrodes in the array owing to the fixed length of the cochlea. It is worth emphasizing that

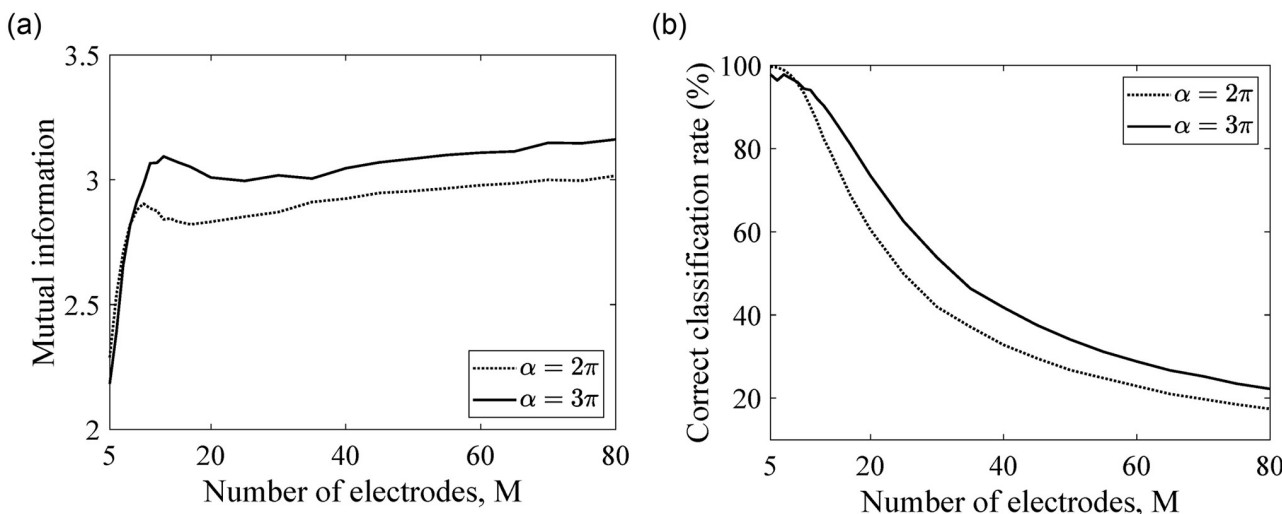

**Fig 4. Model performance vs. the number of electrodes.** Two quantities are shown: (a) mutual information and (b) correct classification rate.

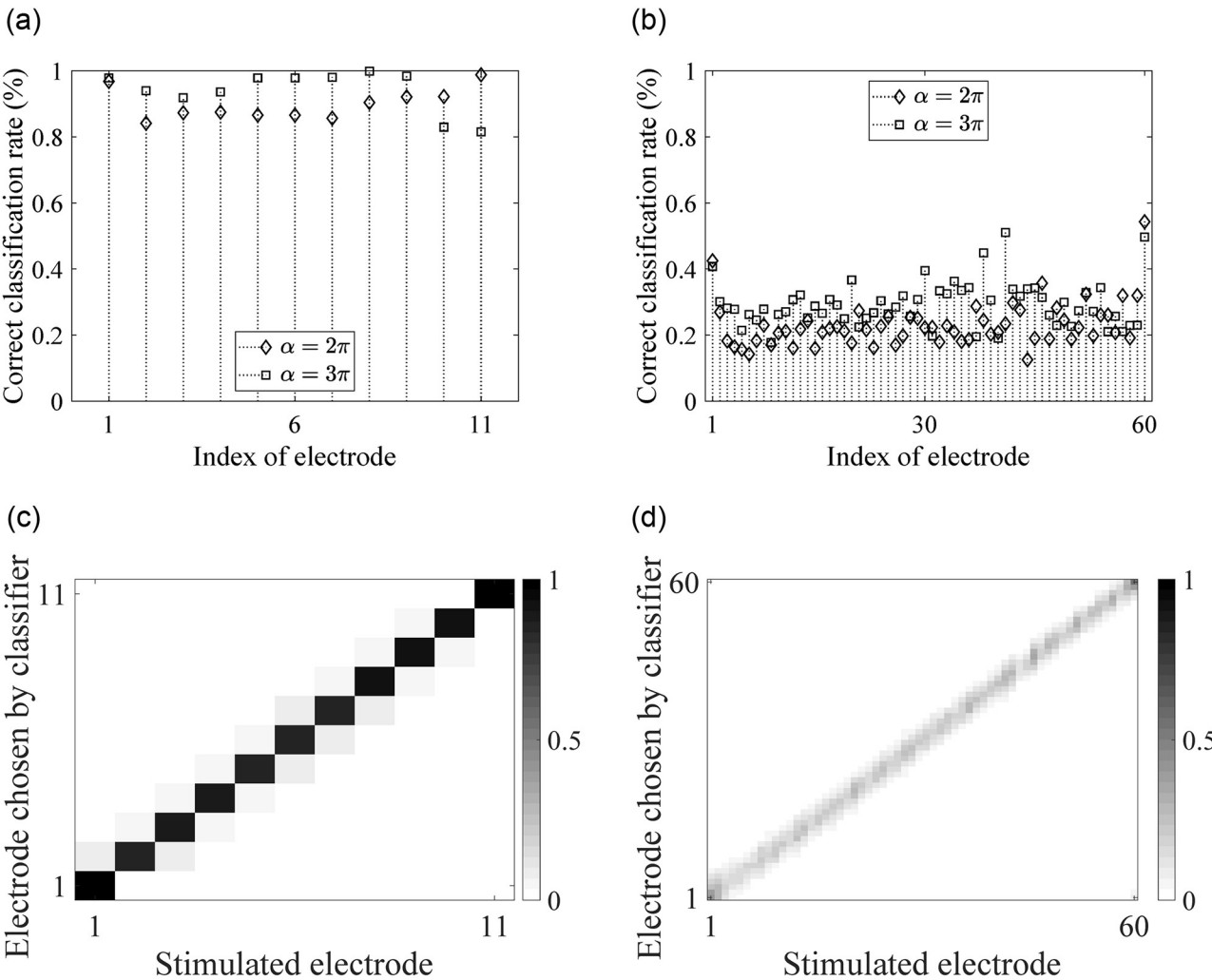

**Fig 5. Electrode discrimination abilities for each electrode in an array.** (a) Correct classification rate for an 11-electrode array, (b) Correct classification rate for an 60-electrode array. (c) Confusion matrix for the 11-electrode array for twirling angle $\alpha = 2\pi$. (d) Confusion matrix for the 60-electrode array for twirling angle $\alpha = 2\pi$. Color bars indicate percentages for each stimulated-chosen electrode pair. We number the electrodes from the basal end to the apical end in sequence.

improvement of correct classification rate for a longer electrode array is not only related to the spacing of adjacent electrodes. We have shown that, with the same spacing between adjacent electrodes, the longer electrode array leads to a better correct classification rate than a shorter electrode array [42].

In our model, correct classification rates quantify the probabilities that each electrode can be correctly discriminated, whereas mutual information is the measurement of the "information content" between a stimulated electrode and the response of auditory nerve fibers. Therefore, increased mutual information is usually associated with improved speech perception, and the maximum mutual information can be associated with the number of electrodes that leads to optimal hearing perception.

To further investigate the how mutual information is related to classification rate, we compare the correct classification rate at each electrode between an 11-electrode array and a 60-electrode array shown in Fig 5(a) and 5(b), respectively. For the two chosen electrode

arrays, there is only a small difference in mutual information as seen in Fig 4(a). However, the electrode discrimination abilities are very different between the two arrays.

Fig 5(c) and 5(d) show electrode discrimination confusion matrices for the 11-electrode and 60-electrode arrays for twirling angle $\alpha = 2\pi$, respectively. For each stimulated electrode, we observe interference between the neighboring 2 electrodes for the 11-electrode array and between the neighboring 4 to 6 electrodes in the 60-electrode array. This indicates that around 12 electrode locations (in which case, the distance between adjacent electrodes is approximately 2.5 mm) can be distinguished. This concords with the result that mutual information calculation. The mutual information for the 11-electrode and 60-electrode arrays are approximately the same, which indicates that the level of perception of 11-electrode and 60-electrode arrays are similar. More importantly, only a marginal increase of mutual information is observed for electrode arrays with more than 12 electrodes for the given parameters in the model, which indicates that the hearing perception does not significantly improve for more than 12 electrodes. This finding is consistent with clinical studies [49, 50].

## Parameter sensitivity

We now model how the key parameters affect the performance of the model. The electrode placements (modeled by $r$), the surviving rate of auditory nerve fibers ($N$), the level of electrode current ($C_e$, which is modeled as the percentage of surviving auditory nerve fibers $N$ that generate action potentials for a given stimulated electrode), and the electrode stimulation strategies (modeled by the attenuation of the current, $A$) are investigated in this parameter sensitivity test.

We choose $\alpha = 2\pi$, $N = 3000$, $r = 2$ mm, and 10% of the surviving auditory nerve fibers generate action potentials for a given stimulated electrode as a general case for a parameter sensitivity test. In each test, we vary one parameter at a time. We use the percentage of surviving auditory nerve fibers, $N$, that generate action potentials for a given stimulated electrode to represent the changes of electrode current, $C_e$.

Fig 6 shows mutual information and correct classification rates for ranges of parameter choices. Together with the results in Fig 4, we investigate the sensitivity of the model to the parameters listed in Table 1. Our framework can clearly reflect impact upon performance with the changes of key parameters. When the change of a chosen parameter leads to a decrease of correct classification rate, a decrease of mutual information is also found. Empirical evidence [11] show that a closer electrode-inner wall distance has a narrower excitation region per electrode and, therefore, leads to better discrimination between two adjacent stimulations. This concords with our results in the first column in Fig 6, which shows that reducing $r$ (reducing the distances of electrodes to nerve fibers) improves performance.

We notice that only small changes are shown when varying electrode-to-fiber distance, $r$, in the model. This is because electrode discrimination is not the ideal test for showing how the electrode-to-fiber distance affects hearing performance. Previous studies have suggested that up to 120 spectral channels can be distinguished via simultaneous virtual channel stimulation [51, 52]. However, speech recognition ability is not significantly improved even if more than 22 individual spectral channels can be distinguished. To better investigate how electrode-to-fiber distance affects hearing performance with cochlear implants, speech recognition should be modelled in the future.

The third column in Fig 6 shows that better performance of the model is achieved by choosing the current level such that $0.2N$ fibers generate action potentials for a given stimulated electrode; i.e., the current level is between 53.8 to 55 $dB$ relative to 1 $\mu A$ for our given parameter set (shown in Fig 7) for an example 25-electrode array.

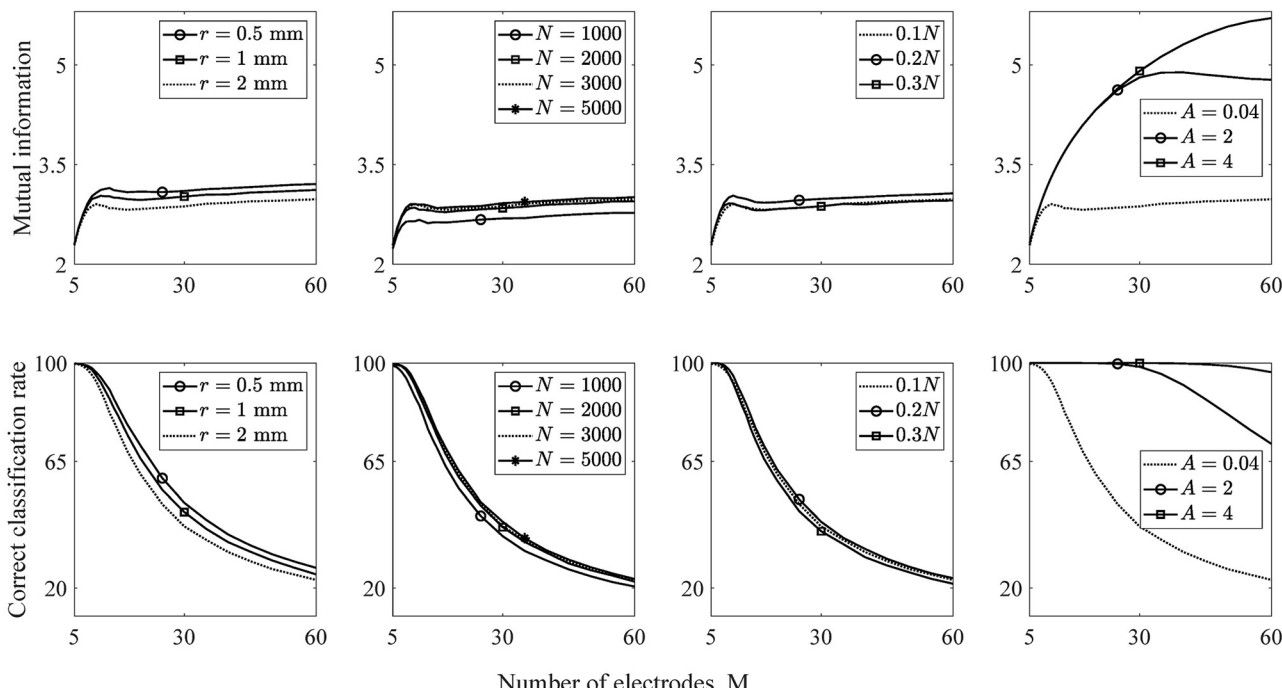

**Fig 6. Model performance with ranges of parameter choices.** The top and bottom rows respectively show mutual information and the percentage of correct electrode discrimination vs. the number of electrodes. The columns from left to right show the sensitivity to $r$, $N$, $C_e$, and $A$, respectively.

The result provides an indication of the "optimal" current level for each electrode. If the current level is too low, there are insufficient fiber activations to provide information about which electrode was stimulated; if current level is too high, there is increased interference between adjacent electrodes.

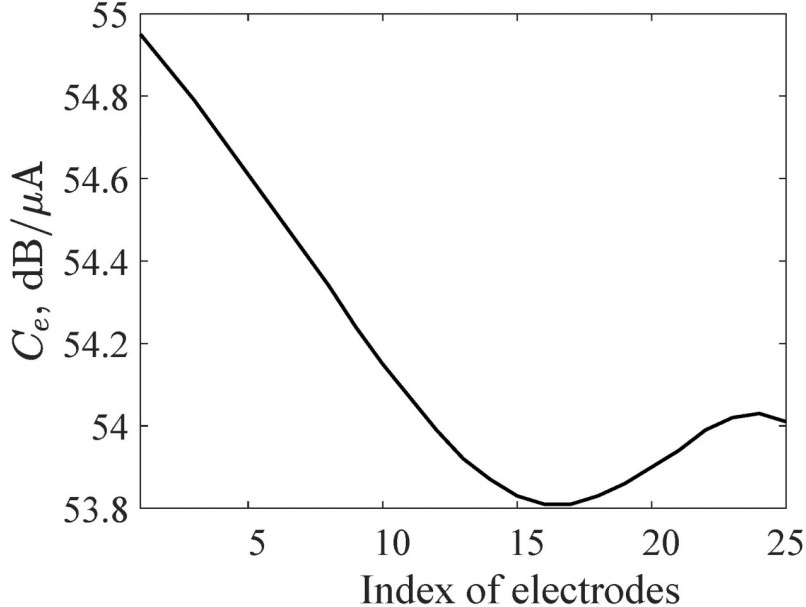

**Fig 7. Current levels, $C_e$, for which 20% of auditory nerve fibers generates action potentials.** A 25-electrode array is used in this example.

The fourth column of Fig 6 shows that increasing *A* (reducing current spread) leads to increase of both mutual information and electrode discrimination ability compared to other parameters. This indicates that better performance of cochlear implant model may be achieved by narrowing the width of current spread such as using current focusing techniques. Our results qualitatively agree with clinical studies; e.g., a current focusing stimulation strategy improves speech recognition in cochlear implant users [24].

## Discussion and conclusion

By combining the information theoretic methods and a machine learning, our model provides a flexible framework that can interrelate cochlear implant electrode discrimination with information transfer at the interface between the electrode array and auditory nerve fibers. Our model also enables us to investigate the extent to which key factors affect the performance of cochlear implant model at the individual level. Our results have shown good qualitative consistency with clinical studies.

### The choice of classifier

A SLFN is chosen as a classifier for electrode discrimination rather than use a simpler classification method as proposed by [35]. The neural network has reduced the limitations of our previous work [35, 42], where the mutual information was based on statistical correlations and relied on an observable output distribution [34, 35]. Choosing a neural network as a classifier provides flexibility to the modeling framework so that it can be adapted to model other psychophysical hearing measurements.

The classifier in this model consists of one hidden layer since electrode discrimination is a simple task. Through trial and error, the optimal classification was found to occur with 1500 hidden layer units and the maximum mutual information occurred with 11 or 12 electrodes. Over-fitting occurred when more than 1500 hidden layer units were used [42]. The activation function in this classifier is the absolute value function; other nonlinear activation functions may also be used effectively [42, 47].

The training and testing data are the activation patterns generated for every possible electrode location, from which 80% of the simulation data was used for training and 20% was used for testing. Although only small variations of the activation patterns exist in the simulation data due to the limited numbers of electrodes, the performance of the classifier was not affected. This is because the neural network classifier only specifically assigns an activation pattern generated by a given electrodes to one of the possible electrode locations rather than act like general classifier to distinguish between activation patterns.

We emphasize that the architecture of the neural network needs to be adjusted according to the complexity of the psychophysical measurements that we aim to simulate.

### Model parameters

Our improved framework successfully demonstrates the impacts of changes to key parameters, and so can provide a theoretic estimation of the performance of cochlear implant model. Thus, the model may be helpful for investigating how and to what extent key factors affect the performance of cochlear implant model for individual users. The key parameters in the model represent the key factors that have been found to influence the performance of cochlear implant model in clinical studies [3, 6, 14]: i) electrode placement can be modeled in the electrode-to-fiber geometry, ii) onset of deafness of cochlear implants users affects the number of surviving auditory nerve fibers and can be modeled by changing the number and

distribution of fibers in the model, and iii) loudness judgements can be modeled by adjusting stimulation current, $C_e$.

The framework enables the investigation of individual components without affecting other components in the framework. In future work, there are a number of ways that the model may be modified to investigate other effects:

1. To test parameter sensitivity, the distance between electrodes and auditory nerve, $r$, was chosen as a uniform distribution. The electrode-cochlea geometry can be updated to reflect actual electrode-to-fiber distances based on CT scans or other measurements [20, 53].

2. To better understand the variability of speech recognition, it is important to model speech recognition tests along with electrode discrimination.

3. The model in this study was built based on "place coding" [39] to study spectral information transfer from electrodes to auditory nerve fibers. A key component to include in the future will be temporal coding, which may improve the accuracy of the model and is necessary to model speech and pitch perception [30, 54].

4. The auditory nerve activity patterns evoked by electrical stimulation and current spread are currently described using a simplified mathematical model. More sophisticated models of current flow in neural tissue [55–58] can be used for a more accurate model predictions.

In this study, we have, for the first time, unified our information theoretic method with a machine learning technique to model electrode discrimination ability, and have overcome the limitations of our previous studies [35, 42]. The flexible framework can be updated to be a subject-specific model of hearing with cochlear implants. This may help with personalized cochlear implant design and provide insight to modeling studies of other types of neural prostheses.

## Supporting information

**S1 Data.**
(7Z)

## Acknowledgments

We thank Dr M. M. A. Yajadda for the valuable discussion and comments on the paper.

## Author Contributions

**Conceptualization:** David Grayden, Mark McDonnell.

**Formal analysis:** Xiao Gao.

**Funding acquisition:** Xiao Gao.

**Investigation:** Xiao Gao.

**Methodology:** Xiao Gao, David Grayden, Mark McDonnell.

**Project administration:** Xiao Gao.

**Supervision:** David Grayden, Mark McDonnell.

**Writing – original draft:** Xiao Gao.

**Writing – review & editing:** Xiao Gao, David Grayden, Mark McDonnell.

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
