## [Decision Letter · Decision Letter 0]

25 Mar 2021

PONE-D-21-01888

Unifying information theory and machine learning in a model of electrode discrimination in cochlear implants

PLOS ONE

Dear Dr. Gao,

Thank you for submitting your manuscript to PLOS ONE. After careful consideration, we feel that it has merit but does not fully meet PLOS ONE’s publication criteria as it currently stands. Therefore, we invite you to submit a revised version of the manuscript that addresses the points raised during the review process.

We look forward to receiving your revised manuscript.

Kind regards,

Andreas Buechner, PhD

Academic Editor

PLOS ONE

Journal Requirements:

1. Please ensure that your manuscript meets PLOS ONE's style requirements, including those for file naming. The PLOS ONE style templates can be found athttps://journals.plos.org/plosone/s/file?id=wjVg/PLOSOne_formatting_sample_main_body.pdf and

Additional Editor Comments (if provided):

Reviewers' comments:

Reviewer's Responses to Questions

**Comments to the Author**

1. Is the manuscript technically sound, and do the data support the conclusions?

Reviewer #1: Yes

Reviewer #2: Partly

2. Has the statistical analysis been performed appropriately and rigorously? 

Reviewer #1: Yes

Reviewer #2: I Don't Know

3. Have the authors made all data underlying the findings in their manuscript fully available?

Reviewer #1: Yes

Reviewer #2: Yes

4. Is the manuscript presented in an intelligible fashion and written in standard English?

Reviewer #1: Yes

Reviewer #2: Yes

5. Review Comments to the Author

Reviewer #1: The authors present a method to model the effect of several of the most important parameters on CI electrode discrimination. In this method they combine an information theory approach that computes the mutual information between input and output with a classifier that predicts the stimulated electrode from a given nerve fiber activation pattern. The method is described clearly and the manuscript is written in an intelligible fashion.

Even though both, the classifier and the information theory approach are described in their own sections, the manuscript would benefit from a more detailed description of the combination of the two methods.

In the results section, the authors could elaborate more on the discrepancies between the results. Why are there substantial differeces between the mutual information and the correct electrode classification rate for the same set of parameters?

Minor remarks:

line 33: the word order seems wrong

line 169: missing word or incorrect grammar

line 192: reference to the wrong figure (should be 4(a))

line 300: "model" instead of "mode" and "have" instead of "has"

Fig. 7: The x axis label does not match the figure description.

Reviewer #2: The manuscript „Unifying information theory and machine learning in a model of electrode discrimination in cochlear implants“ presents an original research article. The model unifying a phenomenological model of auditory nerve activity with a machine learning approach and an information theory approach is very original.

1) Lack of validation of the model. I would suggest that the authors make an effort comparing in more detail the predictions of the model with more detailed data in the literature. For instance one could compare published data on electrode discrimination with different electrode distances and compare it to the predictions of the model. But this is just an example, the authors should make an effort in looking for the exact data that can validate each of the simulations (Figures) of the model. This is a journal paper that extends previous works from the authors, so now it is an opportunity to demonstrate that this model/framework can at least reproduce qualitatively electrode discrimination tasks. For example, if you refer to virtual channel discrimination you should provide data about virtual channel discrimination and demonstrate that your model obtains the same scores as the data.

2) Some mathematical formulation isnot clear. When describing the neural network you say that you train the Wout coefficients, but what happens to the Win coefficients? Are these not trained? If not, why not?

3) Across the manuscript you use the term “performance of cochlear implants”. I would suggest to substitute these terms by “performance of cochlear implant users”.

4) You did a nice sensitivity analysis of your model for some parameters. However, what is the effect of choosing different values for parameters of your network? Size of the hidden layer? Type of activation functions, etc?

Detailed comments

Abstract

“limit performance of cochlear implants”  “Limit performance of cochlear implant users”

“It provide insights it provide insights”  “it provides insights”

Introduction

Third line: What do you mean here with extracellular electrodes? I agree the electrodes are extracellular but this definition seems a bit out of cochlear implant context.

Last line of the first paragraph: Here you give three references at the very end [3,6,7]. Please provide detailed reference to each of your statements i); ii) and iii). Please check if results in the literature are statistically significant for each of the factors.

First line second paragraph: “performance of cochlear implant” � “performance of cochlear implant users”. Please correct this terminology across the whole manuscript where it is used multiple times.

Second line of the second paragraph: Here you state “… electrode dissemination has been used as a primary psychophysical measurement to assess the performance cochlear implants” I disagree with this statement. As far as I know there are no studies showing a significant correlation between electrode discrimination and speech understanding performance. Otherwise please provide these references. I agree though that many researchers investigated electrode discrimination performance as you mention in the following lines, and that this measure may impact speech understanding performance. But it is not a primary measurement, for sure not used in clinical environments.

Page 2 Line 4: Rewrite as “A focus of recent studies is to investigate how …” Actually the two parts of this sentence seem to be a bit disconnected : First part is about “insertion depth” second part is about “virtual channels”. Divide the sentence into two and expand what you are trying to convey here.

Page 2 second paragraph Line 1: “Mathematical and computational model” � “Mathematical and computational models”

Page 2 second paragraph Line 2: Here you cite [15] and [16] but there are many models out there. I would suggest to extend to recent models by Kalkman et al. (2015), Nogueira et al. (2016) and Bai et al. (2019).

Page 2 second paragraph Line 6: What is a Psychophysical cochlear implant model? Please define.

Page 2 second paragraph Line 12: “[25] demonstrated …” Here you can refer to more recent models by Jürgens et al 2018 PlosOne.

Page 3 Caption of Figure 1: Specify the units after 5pi

Page 3 Equation 3: How accurate is this simple model of auditory nerve spiking? For example how accurate can you reproduce auditory nerve activity in comparison to data or to other models such as Joshi or Litvak.

Page 4 second last line of the last paragraph “predication” should be “prediction”?

Page 5 Line 1: “…we defined a vector, T, where each …” instead of a vector, isn’t T a matrix?

Page 5 second line after Equation 7. I would suggest to remove “neuron” and just use the term “activation function” to make sure that the rather does not conus the activation function with Equation 3.

Page 5 first line after equation 8: Here you state that Wout is obtained by training. What happens with Win?

Which activation function did you use in the output layer of your network. I think it is not specified.

Results

Page 6 first line: Here again use “cochlear implant users”

Page 6 last line first paragraph: “400 tests” How many for each electrode?

Page 6: “we find higher correct classification rate for a longer electrode array (alpha = 3pi). Is this improvement caused by the electrode array being longer or just by the fact that the electrode spacing is longer? In other words, you should show that a short array with increased electrode spacing results in worse performance.

Page 7: Caption of Figure 5: Define electrode index.

Page 8: First line second paragraph: “We notice that only small changes are shown when varying electrode-to fiber.. .. this is because electrode discrimination is not as sensitive as speech recognition to changes in r” This statement seems a bit strange. You should provide references and give more detail why this happens. If r is increased electric spread will increase? Why are you not observing a worsening in electrode discrimination? Maybe you can check data comparing electrode discrimination with modiolar vs lateral wall electrodes? Or electrode discrimination with positioner vs no-positioner. Your model needs more data to be validated in general, at least qualitatively.

Page 8 second paragraph last line: “previous studies have suggested 120 spectral channels..” I don’t understand the connection between this sentence and the previous ones. In you model you did not model virtual channels. Please model virtual channels or be more specific with the data that you use to validate your model.

Page 8: Last line of the second last paragraph: Over-training or Over-fitting?

Page 9: Last paragraph: I should be “to model electrode … “ In this section again use “performance of cochlear implant users“

I repeat: In general in the results section I miss more comparison to studies in humans comparing your model predictions with real data, at least qualitatively.

6. PLOS authors have the option to publish the peer review history of their article (what does this mean?). If published, this will include your full peer review and any attached files.

Reviewer #1: No

Reviewer #2: No

---

## [Author Response · Author response to Decision Letter 0]

13 Jun 2021

Dear Editors,

We thank the referee for their overall positive feedback to our manuscript (PONE-D-21-01888). The discussion has been very useful as it has identified ways in which we needed to improve the descriptions in the manuscript. We describe how we have changed the paper to address the reviewers’ comments in the Response to Reviewers.

We believe the paper now has addressed the reviewers’ comments and has improved to better fit the journal requirements.

---

## [Decision Letter · Decision Letter 1]

7 Sep 2021

Unifying information theory and machine learning in a model of electrode discrimination in cochlear implants

PONE-D-21-01888R1

Dear Dr. Gao,

We’re pleased to inform you that your manuscript has been judged scientifically suitable for publication and will be formally accepted for publication once it meets all outstanding technical requirements.

Kind regards,

Andreas Buechner, PhD

Academic Editor

PLOS ONE

Additional Editor Comments (optional):

Reviewers' comments:

Reviewer's Responses to Questions

**Comments to the Author**

1. If the authors have adequately addressed your comments raised in a previous round of review and you feel that this manuscript is now acceptable for publication, you may indicate that here to bypass the “Comments to the Author” section, enter your conflict of interest statement in the “Confidential to Editor” section, and submit your "Accept" recommendation.

Reviewer #1: All comments have been addressed

Reviewer #2: All comments have been addressed

2. Is the manuscript technically sound, and do the data support the conclusions?

Reviewer #1: Yes

Reviewer #2: Yes

3. Has the statistical analysis been performed appropriately and rigorously? 

Reviewer #1: Yes

Reviewer #2: Yes

4. Have the authors made all data underlying the findings in their manuscript fully available?

Reviewer #1: Yes

Reviewer #2: Yes

5. Is the manuscript presented in an intelligible fashion and written in standard English?

Reviewer #1: Yes

Reviewer #2: Yes

6. Review Comments to the Author

Reviewer #1: The authors have appropriately responded to all comments on the previous draft of their paper and have adressed all the issues raised.

Reviewer #2: the manuscript has much improved during the first review and should now be accepted - no further comments or questions from my side.

7. PLOS authors have the option to publish the peer review history of their article (what does this mean?). If published, this will include your full peer review and any attached files.

Reviewer #1: No

Reviewer #2: **Yes: **Waldo Nogueira

---

## [Editor Report · Acceptance letter]

10 Sep 2021

PONE-D-21-01888R1 

Unifying information theory and machine learning in a model of electrode discrimination in cochlear implants 

Dear Dr. Gao:

I'm pleased to inform you that your manuscript has been deemed suitable for publication in PLOS ONE. Congratulations! Your manuscript is now with our production department. 

Kind regards, 

on behalf of

Professor Andreas Buechner 

Academic Editor

PLOS ONE